# Study protocol for evaluating the effectiveness of depression management on gylcaemic control in non-communicable diseases clinics in Malawi

Michael Mphatso Udedi,[1,2,3] Brian W Pence,[4] Felix Kauye,[1,3] Adamson S Muula[3,5]

[1]Department of Mental Health, College of Medicine, University of Malawi, Blantyre, Malawi
[2]Department of Clinical Services, Ministry of Health, Lilongwe, Malawi
[3]Department of Public Health, School of Public Health and Family Medicine, College of Medicine, University of Malawi, Blantyre, Malawi
[4]Department of Epidemiology, University of North Carolina at Chapel Hill, Chapel Hill, North Carolina, USA
[5]Africa Center of Excellence in Public Health and Herbal Medicine, College of Medicine, University of Malawi, Blantyre, Malawi

**Correspondence to**
Michael Mphatso Udedi;
mphatsoudedi@yahoo.co.uk

## ABSTRACT

**Introduction** Depression is associated with negative patient outcomes for chronic diseases and likely affects consistent physical non-communicable diseases (NCDs) care management in relation to clinic attendance and medication adherence. We found no published studies on the integration of depression management in physical NCD clinics in Malawi and assessing its effects on patient and service outcomes. Therefore, the aim of this study is to evaluate the effectiveness of integrating depression screening and management in physical NCD routine care on patient and service outcomes in Malawi. We will also determine the sensitivity and specificity of the Patient Health Questionnaire-9 (PHQ-9) in the detection of depression in NCD clinics.

**Methods and analysis** The study will have two phases. Phase I will involve the validation of the PHQ-9 screening tool for depression, using a cross-sectional study design involving 323 participants, in two specialised physical NCD clinics in one of the 28 districts of Malawi. Using a quasi-experimental study design in four districts of Malawi not involved in the phase I study, the phase II study will evaluate the effectiveness of integrating depression screening (using PHQ-9) and management (based on a specially designed toolkit). Outcomes will be measured at 3 months and 6 months among patients with comorbid diabetes (poorly controlled) and depression attending physical NCD clinics in Malawi.

**Ethics and dissemination** Ethical approval was obtained from the University of Malawi, College of Medicine Research and Ethics Committee (COMREC) on 31 August 2017 (reference P.07/17/2218). The findings will be disseminated through presentations at journal clubs, senior management of the Ministry of Health, national and international conferences as well as submission to peer-reviewed publications. Policy briefs will also be created.

**Trial registration number** PACTR201807135104799.

## Strengths and limitations of this study

► The cross-sectional study design in phase I will allow us to estimate the prevalence of depression among people with diabetes.
► The quasi-experimental design will enable us to investigate the impact of depression in determining the outcome of diabetes.
► The quasi-experimental design which will be used in the second phase has a limitation that baseline differences between the clinics may lead to spurious results.

## INTRODUCTION

Physical non-communicable diseases (NCDs) and depression individually are common and significant causes of morbidity, disability and mortality in sub-Saharan Africa.[1] NCDs account for a growing burden on the population and health systems in low-income and middle-income countries (LMICs).[2] Similarly depression is projected to become a leading cause of disability by 2030.[3] The World Health Survey of 2007 suggests that the prevalence of comorbidity of physical NCDs and depression is increasing; thus between 9.3% and 23.0% of people with NCDs had comorbid depression worldwide.[4] The global prevalence of major depression in people with diabetes is 27%.[5] The relationship between depression and physical NCDs is bidirectional. Modifiable risk factors for physical NCDs are exacerbated by poor mental health and on the other hand physical NCDs are a risk factor for mental illness.[6] Comorbidity is associated with a range of outcomes, including poor adherence or non-adherence to medication, functional impairment, increased medical costs, poor self-care regimens, increased medical symptom burden, and increased morbidity and mortality.[7–9] The recent international study involving 14 countries shows that people with diabetes have depressive disorder as well as significant levels of depressive symptoms. Management of depression (pharmacological treatment combined with psychotherapy) in diabetes mellitus has been shown to be effective both on depression outcomes as well as diabetes management and glycaemic control.[10 11]

Physical NCDs are common and in Malawi, it is estimated that NCDs account for at least 12% of total disability-adjusted life years.[12] In addition, NCDs are the second leading cause of deaths in adults after AIDS.[12] Furthermore, a 2009 nationwide STEPwise approach to Surveillance Survey showed that about 6% of the general population had diabetes mellitus.[13]

Depression commonly overlaps with physical NCDs and depression complicates NCD care. We are not aware of any studies done in Malawi that assessed the burden of depression in physical NCD care and its effect on diabetes outcomes. A recent study done in one of the 28 districts of Malawi reported that control of diabetes is poor and there was high non-adherence of NCD clients to appointments.[14] Currently there are no published studies done in Malawi to explore factors associated with poor control of diabetes and the high rate of non-adherence to attending appointments in physical NCD clinics.

In Malawi, there are few studies looking at depression screening and treatment in general care. The prevalence of depression among outpatient attendees in primary care clinics in Malawi in 2010 and 2013 ranged between 19% to 30%.[15 16] Despite the high prevalence of depression in general healthcare services in Malawi, the extent of comorbid depression in the context of NCDs is unclear. Using the results from general primary care clinic studies done in Malawi on detection rates for depression, it is likely that detection rates of depression in physical NCD clinics in Malawi are poor. Undiagnosed and untreated depression more likely contributes to poor control of diabetes as shown in studies done in other countries.[17–20] Evidence from a recent international study shows that identification and treatment of mental health problems is not routine in clinical care.[20] In Malawi, routine screening will assist in early detection of patients with comorbid depression in physical NCD clinics and facilitate treatment. To our knowledge, there are no studies done in Malawi focusing on identification and interventions for depression in physical NCD clinics. The integration of mental healthcare into other health services is one of the strategies in the National Mental Health Policy in Malawi.[21] The results of the proposed study will help to inform on decisions whether to integrate the strategies for depression management in physical NCD clinics in Malawi.

## AIMS AND OBJECTIVES

Our overall objective is to evaluate the effectiveness of integrating screening and depression management into physical NCD routine care on patient and service outcomes in Malawi. In this way, we will contribute to evidence on comorbidity of depression and diabetes including the effects of identifying and treating depression in patients with comorbid diabetes and depression in physical NCD care in Malawi.

Our study has four specific objectives:

1. To determine the sensitivity and specificity of the Patient Health Questionnaire-9 (PHQ-9) in the detection of depression in NCD clinics.
2. To estimate the prevalence of comorbidity between depression and diabetes mellitus in Malawi.
3. To determine detection rate of depression in physical NCD clinics in Malawi.
4. To compare patient outcomes in physical NCD clinics with and without integration of depression screening and management.

## RESEARCH QUESTIONS

The main research questions in this study are:
1. Will the validated Chichewa PHQ-9 demonstrate satisfactory sensitivity and specificity in physical NCD clinic settings in Malawi?
2. Will integrating depression screening using a validated PHQ-9 and depression management (based on a specially designed toolkit) be more effective than usual care in identifying patients with comorbid depression and diabetes, facilitating treatment of depression in patients with comorbid depression and diabetes, and improving clinical outcomes among patients with comorbid depression and diabetes?

## METHODS AND ANALYSIS

The study will be conducted from December 2017 to December 2018. The study will be in two phases. Phase I will be for validation of PHQ-9 among diabetes patients. Phase II will be evaluating the effectiveness of integrating screening and depression management in patients with comorbid depression and diabetes attending physical NCD clinics. To answer the research questions, we will use a cross-sectional study design in phase I and a quasi-experimental study design in phase II.

### Patient and public involvement

Patients and or public were not involved in the design of this study; however the results will be disseminated to study participants through the clinics' diabetes support groups.

### Phase I: Validation of PHQ-9 in a physical NCD clinic setting in Malawi

#### Study design, area and physical NCD clinics

Depression varies based on cultural context as such screening tools must be adapted and validated for particular populations. PHQ-9 has not been validated in Malawi hence the study will assess validity of PHQ-9. For phase I, we will use a cross-sectional study design. The study will be conducted in Lilongwe district which is one of the 28 districts in Malawi. Malawi is divided administratively into three regions namely Northern, Central and Southern. Lilongwe is located in the central region of Malawi. It has a population of 2688 379 people[22] with eight physical NCD clinics.

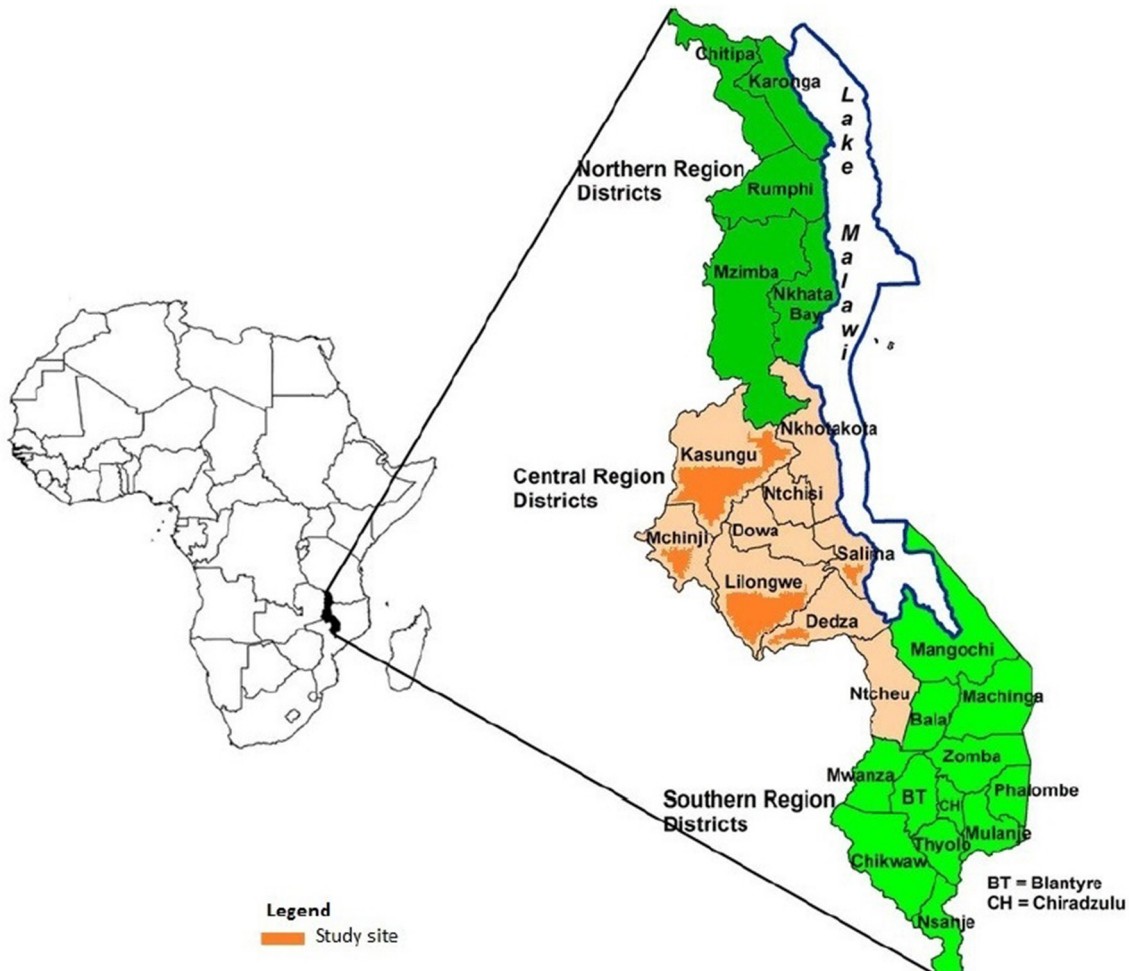

**Figure 1** Map of Malawi showing the central region districts and study sites.

The study will be conducted in two physical NCD clinics (Area 25 Health Centre under Lilongwe District Health Office and Kamuzu Central Hospital). The Area 25 Health Centre serves a catchment population of 80 000 while the catchment population of Kamuzu Central Hospital (a tertiary health facility) is the entire central region which is 7401 068.[22] The physical NCD clinic at Area 25 Health Centre is staffed by clinical officers and nurses who have undergone appropriate training in the diagnosis and management of key NCDs while the clinic of Kamuzu Central Hospital is also staffed by medical officers (general medical doctors) and specialists on top of clinical officers and nurses. The two sites are implementing the WHO Package of Essential NCD (WHO PEN) interventions. Figure 1 is a map of Malawi showing the central region districts.

## SAMPLE SIZE DETERMINATION

The sample size was calculated based on the sensitivity of PHQ-9 using Buderer's formula.[23] We used the following parameters; anticipated sensitivity was 80%, standard normal deviation corresponding to the specified size of the critical region ($z^2_{1-\alpha/2}$) is 3.84, the size of the critical region, alpha ($\alpha$), is 0.05, z is corresponding to the

level of significance and is 1.96, L is the absolute precision desired on either side of sensitivity and was set at 0.1 and prevalence for depression was 20%. The prevalence of depression of 20% is based on a recent study on patients attending a healthcare setting in Malawi.[15] We also considered potential participants' refusal and loss of data; consequently we put it at 5%. Taking these assumptions into consideration, the required sample size was calculated as 323.

## RECRUITMENT AND DATA COLLECTION

Consecutive patients attending the NCD clinic will be identified by research assistants. The research assistants will provide information about the study to the patients and seek informed written consent. Participants will be eligible for the study if they are at least 18 years of age and are attending the NCD clinics for diabetes care and are available for an interview. Participants will be excluded if they require acute medical care or are unable to speak.

The research assistants will enrol patients who consent and carry out interviews at the health facility. When patients do not consent to participate in the study, the research assistant will request permission to record sex, age, level of education and occupation of the patient.

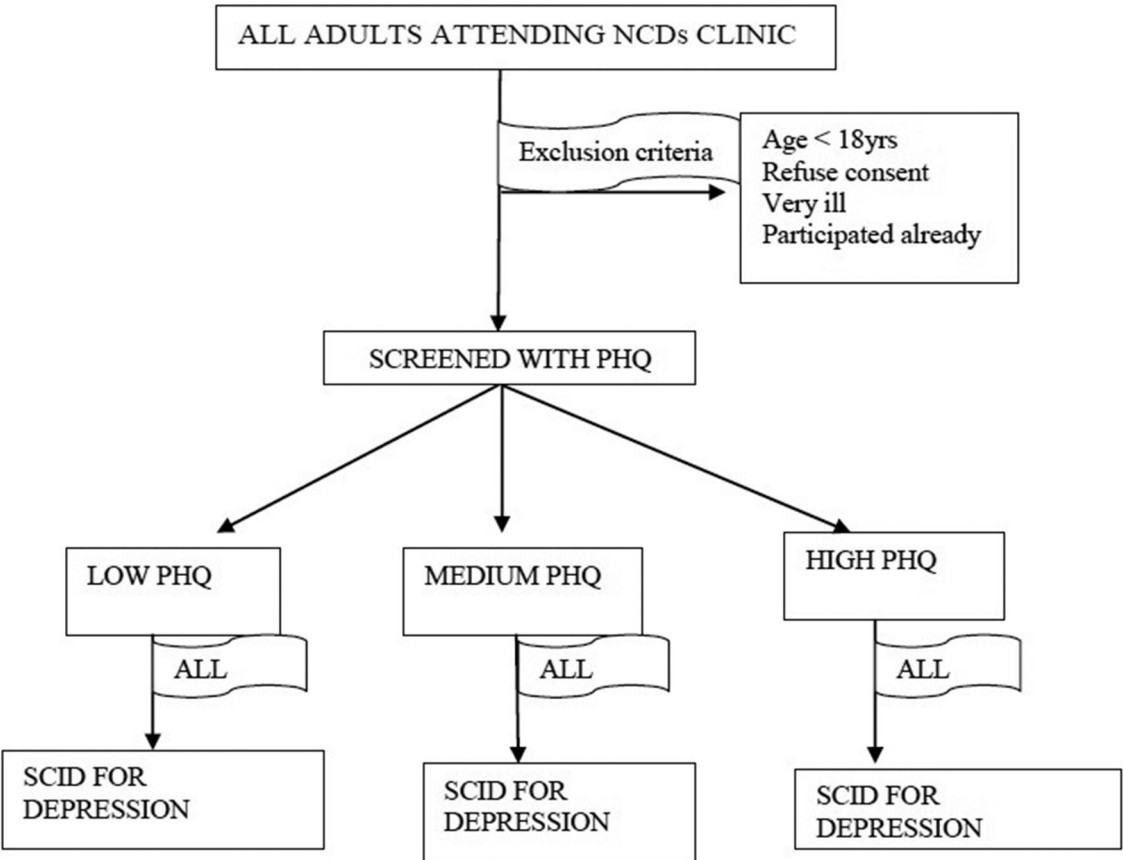

**Figure 2** Phase I study procedure. NCD, non-communicable disease; PHQ, Patient Health Questionnaire.

One research assistant will administer PHQ-9 orally to ensure high and standard responses due to low literacy of the study participants. The research assistant will then calculate the screening score and let the participant be attended to by the NCD clinician. After the consultation the patient will have the Diagnostic and Statistical Manual of Mental Disorders (DSM–IV), Axis I disorders Clinical Version Structured Clinical Interview for DSM (SCID) administered by a second research assistant to elicit the psychiatric symptoms itemised in the DSM criteria for depression. The second research assistant will be masked from the PHQ-9 Score. The training, supervision and quality assurance of the research assistant doing the SCID will be done as well. A diagrammatic representation of the process is shown in figure 2.

## MEASUREMENTS

The aim of phase I is to validate PHQ-9 in physical NCD clinics in order for it to be used for screening of depression in NCDs in Malawi. PHQ-9 (based on DSM criteria) has been chosen for this study because it is brief, requires a short time to administer and is compatible with the Malawian health setting workload. Furthermore, the PHQ-9 can be used to establish depressive disorder diagnoses as well as depressive symptoms' severity.[24]

For screening data using PHQ-9, presence of depressive symptoms will be defined as having a total score of 5 and above on all the nine questions (PHQ-9 total score ≥5). Symptoms will be categorised using established cut-off points 5, 10 and 20 (mild, moderate and severe depressive symptoms, respectively).[25 26] The sensitivity and specificity at the standard cut-off point will be assessed, and performance at an alternative cut-off point will also be examined to determine whether the standard cut-off point is optimal in this population

SCID for depression will be used as the gold standard to validate PHQ-9. SCID for depression has been translated into Chichewa and has been used in Malawi after undergoing a process of validation which included translation, back translation and testing.[27]

### Data management

Data will be checked for consistency and completeness by the principal investigator and double entered to SPSS V.20 by experienced data entry clerks. Then, it will be cleaned and analysed using the same V.20 of SPSS. Hard copies of the data will be stored in a locked cabinet and consent forms will be separated from the data.

### Data analysis

Data will be analysed using SPSS V.20. Descriptive statistics will be computed to describe the sociodemographic characteristics of participants and to summarise the frequencies of depression under each category in both criteria (PHQ-9 and SCID). We will assess whether refusal

is associated with sociodemographic characteristics by testing the statistical significance of sociodemographic characteristics between refusers and consenters. The prevalence of diagnosed depression among patients with diabetes will be determined by computing the proportion of those with depression according to SCID; these patients will be correctly screened out by PHQ.

The sensitivity, specificity and predictive values of a PHQ-9 total score ≥5, with 95% CIs, will be calculated relative to the reference standard of a diagnosis from the depression module of SCID. A receiver operating characteristic curve (ROC curve) will be made using SPSS V.20 in order to get criterion validity of PHQ-9. Internal consistency will be determined by Cronbach's α. In exploratory analyses we will examine the test characteristics of alternative thresholds.

## Phase II: evaluation of the effectiveness of integrating screening using validated PHQ-9 and depression management in physical NCD clinics

### Study design and setting

In phase II of the study, we will use a quasi-experimental design employing a non-equivalent control group (prepost) design. Prospective controlled interventional trials provide a higher level of evidence for a true causality.[28] The best choice of the controlled interventional studies is the true experimental design of which RCT is the gold standard. However a non-randomised control group trial is a reasonable choice when an RCT would be ideal but not feasible due to costs and ethics limitations and also when trying to establish the effectiveness of large-scale dissemination and implementation.[28 29] Furthermore, literature indicates that the interventions often cannot be randomised for the following reasons: ethical considerations, an inability to randomise patients, an inability to randomise locations or settings, and a need to intervene quickly.[29] We have therefore chosen to use a quasi-experimental design and the intervention programme will be done in two district NCD clinics while the other two districts will provide usual care. Four convenient physical NCD clinics which are implementing the WHO PEN interventions will be allocated to intervention arm (two physical NCD clinics) and control arm (two physical NCD clinics) and in each clinic. Consecutive patients who meet the study criteria will be recruited.

The phase II of the study will be conducted for a period of 9–12 months in four healthcare facilities in Mchinji district, Kasungu district, Salima district and Dedza district of the central region of Malawi (figure 1). The four districts are also implementing the WHO PEN interventions. Currently each district has one physical NCD clinic based at the district hospital. The physical NCD clinics in the four districts are staffed by clinical officers and nurses who have undergone appropriate training in the diagnosis and management of key physical NCDs. They follow similar standard guidelines in the management of physical NCDs.

### Intervention

1. Clinicians at the physical NCD clinics will be trained in routine screening for depression with a validated PHQ-9.
2. Clinicians at the physical NCD clinics will be trained on comorbidity of diabetes and depression and on coming up with individual care plans and prescribing and monitoring antidepressants.
3. Clinicians at the physical NCD clinics will be trained on problem-solving skills using effective helping techniques for depression and on how to follow-up individual care plans.

### Usual care

This would involve the NCD clinicians treating patients with comorbid depression and diabetes using normal practices of treating depression involving antidepressants.

## SAMPLE SIZE DETERMINATION

The sample size was based on detecting a difference in glycated haemoglobin ($HbA_1c$) at 6 months between the study arms. It was based on an analysis of covariance (ANCOVA) approach, with the outcome measurement at follow-up, and the baseline $HbA_1c$ considered as a covariate in the analysis. The correlation between baseline and follow-up haemoglobin values is estimated to be 0.5. A difference in $HbA_1c$ between arms of 0.7% is considered to be a clinically important difference (based on a previous study which found a difference of 0.56%,[30]) and data from previous research suggested that the $HbA_1c$ measurements had an SD of 1.8%.[30] It is calculated that using a 5% significance level and 80% power; 78 subjects per group are required. It is estimated that there will be a 10% dropout/loss to follow-up. To allow for this, we planned to recruit 87 subjects per group, giving a total sample size for the study of 174 participants.

## RECRUITMENT AND DATA COLLECTION

We will use a mixed-methods (quantitative and qualitative) approach in phase II. The reason for collecting both quantitative and qualitative data is to complement and also to address the shortfalls of either approach. The use of both forms of data allows researchers to gain deeper understanding of the phenomenon under study.[31] In addition, evidence suggests that triangulation of methods increases both the validity and reliability of data.[32]

The research assistant with the help of health professionals running the NCD clinic at each site will identify the study participants who meet the inclusion criteria consecutively. The research assistants will provide information about the study to the patient and seek written informed consent. Participants will be eligible for the study if they are at least 18 years of age, are attending the NCD clinic for diabetes care and are available for an interview. Participants will be excluded if they require acute medical care or are unable to speak. The research

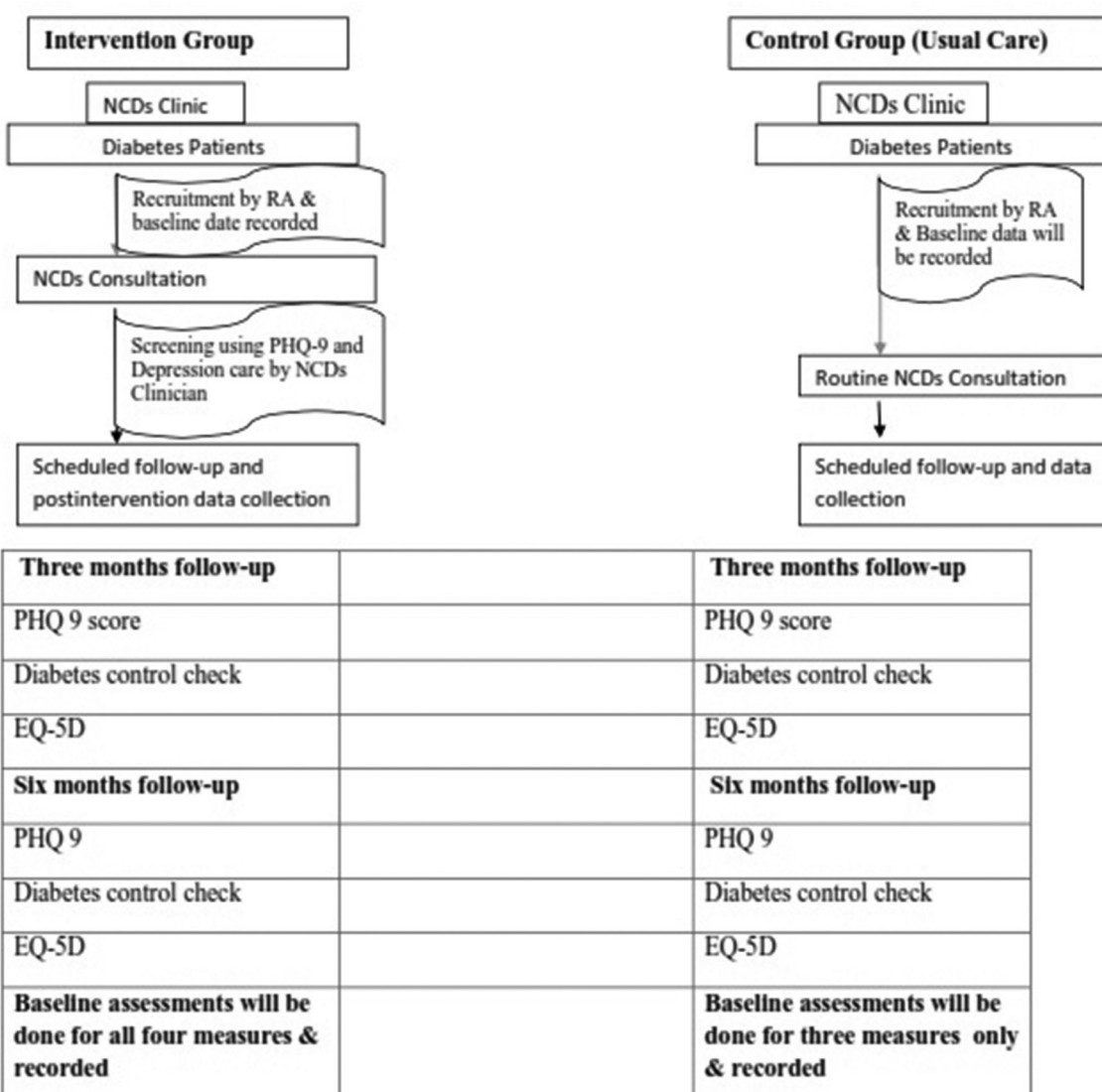

**Figure 3** Phase II study procedure. EQ-5D, EuroQol Five Dimensions Questionnaire; NCD, non-communicable disease; PHQ-9, Patient Health Questionnaire -9.

assistants will enrol patients who consent and carry out interviews at the health facility. When patients do not consent to participate in the study, the research assistant will request permission to record sex, age, level of education and occupation of the patient. Adult patients with poorly controlled diabetes (a HbA$_1$c level of 8.5% or higher as suggested from previous research[30 33]) and comorbid depression will be recruited for the study and followed up to 6 months after recruitment (figure 3). Data collection will occur at three time points: baseline, at 3 months and at 6 months.

Qualitative interviews will be conducted among healthcare workers in order to assess feasibility and acceptability of integrating depression management into NCD care. In-depth interviews will be employed to understand health professionals' experience in using the validated PHQ-9 for screening and the depression management guidelines.

## MEASUREMENTS

### The primary exposure variable

Depression will be measured using PHQ-9. PHQ-9 will have been validated in phase I of this study.

The primary exposure of interest is whether the patient is receiving care at an intervention or usual care facility.

### Outcome variables

#### Primary outcome variable

The patient diabetes outcome for this study is blood glucose control. Data on the patients' status on the primary outcome will be extracted from the NCD patient mastercard. For the purpose of the study in checking how well diabetes is being controlled, HbA$_1$c test will be performed on the recruited study participants at baseline, 3 months and 6 months. Depression is the co-primary outcome. PHQ-9 will be administered at baseline, 3 months and 6 months.

### Secondary outcome variable

Quality of life (QoL) will be measured using the interviewer-administered version of the EuroQol Five Dimensions (EQ-5D) Questionnaire. This instrument has been validated in South Africa and Zimbabwe.[34 35] The tool has been translated into several languages including Afrikaners, Xhosa and Shona.[35] The EQ-5D Questionnaire will be adapted in Malawi and will be used at baseline and follow-up of patients at 3 months and 6 months.

### Data management

Data will be checked for consistency and completeness by the principal investigator and double entered to SPSS V.20 by experienced data entry clerks. Hard copies of the data will be stored in a locked cabinet and consent forms will be separated from the data.

### Data analysis

The primary outcome is $HbA_1c$ at 6 months. This will be compared between groups using ANCOVA. The haemoglobin value at baseline will be considered as a covariate in the analysis.

The $HbA_1c$ values at 3 months will be considered as a separate outcome, and analysed using equivalent methods to the primary outcome.

Quantitative data will be analysed statistically using SPSS V.20. Descriptive analysis will be done. The recorded qualitative data on the feasibility and acceptability of integrating depression management will be transcribed and coded using Atlas.ti. Relevant queries will be created in the database to thematically analyse the data.

### Ethical considerations

Written informed consent will be obtained from every participant and fingerprint impressions will be taken from consenting illiterate participants. Patients identified as having depression or endorsing the suicide item on PHQ-9 will be referred to appropriate professionals for further assessment and management. Permission to conduct the study has been sought from the the district health officers of Lilongwe, Kasungu, Dedza, Mchinji, Salima and the hospital director of Kamuzu Central Hospital. Study participants will not be given any financial reimbursement, however they will thanked verbally for their participation. Privacy will be maintained as the interviews will be conducted in a room which will provide both visual and audio privacy. Referrals will be made to appropriate professionals for further assessment and management of cases where participants will be identified to be under depression as well as cases that will screen positive for suicidal ideations through the interviews.

### Dissemination plan

The findings will be disseminated through publications in peer-reviewed journals and conference presentations. Summary reports will be submitted to the participating institutions and to the University of Malawi, College of Medicine. The results will also be shared with policy makers and other local stakeholders in the form of policy briefs as part of evidence on integration of depression management in NCDs.

### Limitations of the study

In this study, people may have undiagnosed comorbid illnesses and our method of screening may not be strong in capturing other comorbid illnesses. In the first phase of validation we will use two specialised clinics which may not be representative of the wider population. QoL will be measured using five domains and we will therefore not obtain detailed information on the various dimensions that assess health-related QoL. The quasi-experimental design which will be used in the second phase also has a limitation that baseline differences between the clinics may lead to spurious results.

### Strengths of the study

This study will provide the much needed evidence about the impact of comorbid depression on the outcome of diabetes. The cross-sectional study design will allow us to estimate the prevalence of depression among people with diabetes. The study will enable us to investigate the impact of depression in determining the outcome of diabetes. To our knowledge, this study will be the first in Malawi to examine identification and interventions for depression in physical NCD clinics.

### Expected benefits of the findings

Our findings will contribute to sparse evidence on mental health, diabetes and other NCDs in LMICs. We anticipate that our study findings will raise awareness of comorbidity among clinicians, service providers, policy makers and that it will contribute to clinical practice and policy change regarding the delivery of mental healthcare in NCD care in Malawi.

**Contributors** MMU, BWP, FK and ASM were involved in the conceptualisation of the study. MMU was responsible for drafting the protocol manuscript. BWP, FK and ASM edited the protocol. All authors read and approved the final manuscript.

**Funding** This study protocol presents independent work supported through the DELTAS Africa Initiative (DEL-15-01). The DELTAS Africa Initiative is an independent funding scheme of the African Academy of Sciences' (AAS) Alliance for Accelerating Excellence in Science in Africa (AESA) and supported by the New Partnership for Africa's Development Planning and Coordinating Agency (NEPAD Agency) with funding from the Wellcome Trust (DEL-15-01) and the UK government.

**Disclaimer** The views expressed in this publication are those of the author(s) and not necessarily those of AAS, NEPAD Agency, Wellcome Trust or the UK government.

**Competing interests** None declared.

**Patient consent** Not required.

**Ethics approval** University of Malawi, College of Medicine Research and Ethics Committee (COMREC)

**Provenance and peer review** Not commissioned; externally peer reviewed.

**Data sharing statement** The authors will share data of the original research article with their colleagues for academic purposes only. Currently the authors have not shared any unpublished data for the study.

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
