## [Reviewer comments · BMJ Open]

This paper was submitted to a another journal from BMJ but declined for publication following peer review. The authors addressed the reviewers' comments and submitted the revised paper to BMJ Open. The paper was subsequently accepted for publication at BMJ Open.

(This paper received three reviews from its previous journal but only two reviewers agreed to published their review.)

ARTICLE DETAILS

TITLE (PROVISIONAL)	Study protocol for evaluating the effectiveness of depression management on glycaemic control in non communicable diseases clinics in Malawi
AUTHORS	Udedi, Michael Mphatso; Pence, B; Kauye, F.; Muula, Adamson S.

VERSION 1 – REVIEW

REVIEWER	Frank Snoek VU University Medical Center and Academic Medical Center, Amsterdam, The Netherlands
REVIEW RETURNED	08-Apr-2018

GENERAL COMMENTS	This study protocol addresses a highly relevant topic, aiming to enhance recognition and management of co-morbid depression in patients as part of medical care for NCD in Malawi. I understand the protocol was approved end of August 2017 and wonder about the reason for the time-lag. Is the study not yet started? I have some questions for clarification. - Title: 'diabetes outcomes' does not reflect chosen sole parameter (HbA1c) and probably should be changed to "Glycaemic control"- Assuming we are talking about Type 2 diabetes it would make sense to take other clinical measures/risk factors into account that predict morbidity and mortality: blood pressure, lipids, weight... Please comment.- I would suspect the authors to mention depression as a co-primary outcome. Please comment/explain.- There is very little information on the depression treatment (anti-depressants) offered to patients and how that should affect/improve diabetes control. Can the authors elaborate?- The first phase aims to establish sensitivity and specificity of the PHQ-9 in Malawi patients. The authors provide no compelling reason why this needs to be re-established after previous multiple studies outside Malawi. Cultural/linguistic issues?- SCID as gold standard is a strength, but wonder why not anxiety was also included.- PHQ-9 scores/classes are described as mild, moderate, severe (5, 10, 20 resp.). In figure 2 it says low, medium and high: are these the same?- Are other known mental health problems recorded? how is that
---

	information handled in the analyses - Design: a quasi experimental design was chosen. Why not a cluster-randomised trial which would offer a more rigorous test?
--	---

REVIEWER	Professor Cathy Lloyd The Open University Leading a similar study.
REVIEW RETURNED	16-Apr-2018

GENERAL COMMENTS	This protocol reports a very good study on the impact of depression screening within a diabetes clinical setting. I do have some minor comments and concerns which I feel should be addressed prior to publication: The planned study is significantly the same as the INTERPRET-DD study, whose protocol and baseline findings have now been published. The authors have not acknowledged this which worries me. This should be addressed and the relevant references added. Both papers are published in Diabetic Medicine, one in 2015 (so they should have picked that up at least) and one very recently. First author is Lloyd. In terms of their protocol, it is laid out very well. I wondered why the authors haven't included depression or depressive symptomatology as an outcome measure - if the intervention is designed to improve psychological wellbeing then surely they should. This could be just the PHQ-9 or indeed a clinical assessment if they can do one (this is one place where their study differs from INTERPRET-DD). I wondered why they were doing a clinical assessment of depression - it would be useful for the authors to clearly identify what they mean by 'depression' and what are depressive symptoms (the PHQ-9 only measures symptoms). The limitations should include that they have picked two specialized clinics which may not be representative of the wider population. This doesn't invalidate their study - it is common to have to be very pragmatic about where the research is actually carried out. It would be good for the authors to compare their results with data from other African countries as well as to reflect more broadly on the implications for care in Malawi, but this comes with publication of their findings, which I look forward to. I can't find a Strobe statement.
---

VERSION 1 – AUTHOR RESPONSE

Reviewer: 1

Reviewer Name: Frank Snoek

Institution and Country: VU University Medical Center and Academic Medical Center, Amsterdam, The Netherlands

Please state any competing interests or state 'None declared': None declared

Please leave your comments for the authors below

This study protocol addresses a highly relevant topic, aiming to enhance recognition and management of co-morbid depression in patients as part of medical care for NCD in Malawi.

I understand the protocol was approved end of August 2017 and wonder about the reason for the time-lag. Is the study not yet started? I have some questions for clarification.

Response: Thank you for this point. The study has two phases with the first phase being the validation of the PHQ-9 to be used in phase two. The first phase of the study started on 1st December 2017 and the main study has not started. We have just completed the phase one of the study and data for this phase is being analysed. The time-lag was a result of logistical issues which were being handled by Research Support Centre of the College of Medicine (Malawi) for instance recruitment of Research Assistants.

- Title: 'diabetes outcomes' does not reflect chosen sole parameter (HbA1c) and probably should be changed to 'Glycaemic control'

Response: Thank you for this point. We have revised and the title now reads;
“Study protocol for evaluating the effectiveness of depression management on glycaemic control in non communicable diseases clinics in Malawi”.

- Assuming we are talking about Type 2 diabetes it would make sense to take other clinical measures/risk factors into account that predict morbidity and mortality: blood pressure, lipids, weight... Please comment.

Response: We agree with this comment, and some information will be collected in the general questionnaire such as weight and blood pressure as these are routine clinical measures for the NCDs Clinics. In addition due to limited resources in our setting, it would be difficult to do a lipids screen. The best we can do is a Body Mass Index

- I would suspect the authors to mention depression as a co-primary outcome. Please comment/explain.

Response: This point regarding inclusion of depression as an outcome measure is well taken – we have revised accordingly and included depression as a co-primary outcome

- There is very little information on the depression treatment (anti-depressants) offered to patients and how that should affect/improve diabetes control. Can the authors elaborate?

Response: We have added the information in the introduction
“Management of depression (pharmacological treatment combined with psychotherapy) in diabetes mellitus, has shown to be effective both on depression outcomes as well as diabetes management and glycaemic control”.

- The first phase aims to establish sensitivity and specificity of the PHQ-9 in Malawi patients. The authors provide no compelling reason why this needs to be re-established after previous multiple studies outside Malawi. Cultural/linguistic issues?

Response: Thank you for this point. It is indeed cultural/linguistic issues. We have clarified and the phase one section in methods and analysis reads;
“Depression varies based on cultural context as such screening tools must be adapted and validated for particular populations. PHQ-9 has not been validated in Malawi hence the study will assess validity of the PHQ-9”.

- SCID as gold standard is a strength, but wonder why not anxiety was also included.

Response: We are using the SCID for depression which was already translated and validated in Malawi however we do not have an already validated SCID for anxiety in Malawi. Since this is a PhD study we did not have enough resources to validate the SCID for anxiety. In addition, being a PhD study, there is the time factor to be taken into consideration as well besides resources. Having a large number of outcomes would mean more time and resources needed for data collection. Since the leading author has interest in co-morbidity of non-communicable diseases and mental disorders, anxiety disorders will be a future grant proposal area.

- PHQ-9 scores/classes are described as mild, moderate, severe (5, 10, 20 resp.). In figure 2 it says low, medium and high: are these the same?

Response: Thank you for this point. In this context the categories are the same

- Are other known mental health problems recorded? how is that information handled in the analyses

Response: Thank you for this point. Other known mental health problems are recorded in the general questionnaire and this has been considered in the analysis

- Design: a quasi experimental design was chosen. Why not a cluster-randomised trial which would offer a more rigorous test?

Response: This is an important point. We agree that cluster-randomised trial would offer a more rigorous test. However in the context that this is a PhD study we considered cost and time hence the choice of quasi experimental design. The findings of this study will be a basis for a grant application for a more rigorous test using a cluster-randomised trial.

Reviewer: 2

Reviewer Name: Professor Cathy Lloyd

Institution and Country: The Open University

Please state any competing interests or state 'None declared': Leading a similar study.

Please leave your comments for the authors below

This protocol reports a very good study on the impact of depression screening within a diabetes clinical setting. I do have some minor comments and concerns which I feel should be addressed prior to publication:

The planned study is significantly the same as the INTERPRET-DD study, whose protocol and baseline findings have now been published. The authors have not acknowledged this which worries me. This should be addressed and the relevant references added. Both papers are published in Diabetic Medicine, one in 2015 (so they should have picked that up at least) and one very recently. First author is Lloyd.

Response: We have reviewed and now cited this reference. We have revised the text accordingly in the introduction to reflect this.

In terms of their protocol, it is laid out very well. I wondered why the authors haven't included depression or depressive symptomatology as an outcome measure - if the intervention is designed to improve psychological wellbeing then surely they should. This could be just the PHQ-9 or indeed a

clinical assessment if they can do one (this is one place where their study differs from INTERPRET-DD).

Response: This point regarding inclusion of depression as an outcome measure is well taken – we have revised accordingly and included depression as an outcome measure

I wondered why they were doing a clinical assessment of depression - it would be useful for the authors to clearly identify what they mean by 'depression' and what are depressive symptoms (the PHQ-9 only measures symptoms).

Response: In this context “depression” mean the clinical diagnosis of depression and “depressive symptoms” presence of depressive symptoms will be defined as having a total score of 5 and above on all the nine questions of the PHQ-9

The limitations should include that they have picked two specialized clinics which may not be representative of the wider population. This doesn't invalidate their study - it is common to have to be very pragmatic about where the research is actually carried out.

Response: This is an important point. We have indicated the limitation in the manuscript. “In the first phase of validation we will use two specialized clinics which may not be representative of the wider population”.

It would be good for the authors to compare their results with data from other African countries as well as to reflect more broadly on the implications for care in Malawi, but this comes with publication of their findings, which I look forward to.

I can't find a Strobe statement.

Response: Thank you for your observation. We had checked on the journal requirements; as the paper is a protocol paper, the Journal does not require a copy of the STROBE checklist.